# The invisible costs of obstructive sleep apnea (OSA): Systematic review and cost-of-illness analysis

Ludovica Borsoi[1]*, Patrizio Armeni[1], Gleb Donin[2], Francesco Costa[1], Luigi Ferini-Strambi[3]

1 SDA Bocconi School of Management, Centre for Research on Health and Social Care Management (CERGAS), Milan, Italy, 2 Department of Biomedical Technology, Czech Technical University in Prague, Kladno, Czech Republic, 3 Faculty of Psychology, Vita-Salute San Raffaele University, Milan, Italy

* ludovica.borsoi@unibocconi.it

## Abstract

### Background

Obstructive sleep apnea (OSA) is a risk factor for several diseases and is correlated with other non-medical consequences that increase the disease's clinical and economic burden. However, OSA's impact is highly underestimated, also due to substantial diagnosis gaps.

### Objective

This study aims at assessing the economic burden of OSA in the adult population in Italy by performing a cost-of-illness analysis with a societal perspective. In particular, we aimed at estimating the magnitude of the burden caused by conditions for which OSA is a proven risk factor.

### Methods

A systematic literature review on systematic reviews and meta-analyses, integrated by expert opinion, was performed to identify all clinical and non-clinical conditions significantly influenced by OSA. Using the Population Attributable Fraction methodology, a portion of their prevalence and costs was attributed to OSA. The total economic burden of OSA for the society was estimated by summing the costs of each condition influenced by the disease, the costs due to OSA's diagnosis and treatment and the economic value of quality of life lost due to OSA's undertreatment.

### Results

Twenty-six clinical (e.g., diabetes) and non-clinical (e.g., car accidents) conditions were found to be significantly influenced by OSA, contributing to an economic burden ranging from €10.7 to €32.0 billion/year in Italy. The cost of impaired quality of life due to OSA under-treatment is between €2.8 and €9.0 billion/year. These costs are substantially higher than those currently borne to diagnose and treat OSA (€234 million/year).

**Data Availability Statement:** All relevant data are within the paper and its Supporting information files.

**Funding:** CERGAS SDA Bocconi received an unrestricted grant for research from Philips S.p.A. The funder had no role in study design, data collection and analysis, decision to publish, or preparation of the manuscript.

**Competing interests:** CERGAS SDA Bocconi received an unrestricted grant for research from Philips S.p.A. This does not alter our adherence to PLOS ONE policies on sharing data and materials. Ludovica Borsoi, Patrizio Armeni, Gleb Donin and Francesco Costa have no competing interests to declare. Luigi Ferini-Strambi declares the following competing interests (last 3 years): Philips-Respironics (fee for lectures), Resmed (fee for advisory board). This does not alter our adherence to PLOS ONE policies on sharing data and materials.

## Conclusions

This study demonstrates that the economic burden due to OSA is substantial, also due to low diagnosis and treatment rates. Providing reliable estimates of the economic impact of OSA at a societal level may increase awareness of the disease burden and help to guide evidence-based policies and prioritisation for healthcare, ultimately ensuring appropriate diagnostic and therapeutic pathways for patients.

## Introduction

Obstructive sleep apnea (OSA) is a disorder characterized by episodic cessation of breathing due to repeated upper airway partial (hypopnea) or total (apnea) obstructions [1, 2]. These events lead to activation of the sympathetic nervous system, sleep fragmentation, intermittent hypoxemia and, in the case of syndrome (OSAS), to excessive daytime sleepiness [3, 4]. Diagnosis of OSA usually requires overnight laboratory polysomnography in order to detect the frequency of disordered breathing events [5]. OSA's severity is measured through the number of apnea and hypopnea events per hour of sleep (apnea-hypopnea index, AHI), which can be more than 30 in its severe form [6, 7]. The attention towards the disorder is increasing as several studies found that OSA is a risk factor for a substantial number of clinical conditions in adults, including diabetes [8–10], cancer [11, 12], cardiovascular [13, 14] and cerebrovascular diseases [15–17]. Moreover, OSA and its syndrome are associated with decreased quality of life (QoL) [18–20], impaired work performance [21–23] and increased risk of road traffic accidents [24, 25]. According to several population-based studies, prevalence of OSA is relatively high, especially among men [26–31], although methodological differences and difficulties in characterizing this disorder yielded to variability in prevalence estimates [32, 33]. Despite the prevalence of this condition, OSA is frequently undiagnosed, either because patients do not regard their symptoms (e.g., snoring, excessive daytime sleepiness) as the presence of a disorder, thus not seeking medical consultation, or because primary care physicians are often unable to recognize OSA signs and symptoms [5, 34], leading to a potential underestimation of the disease burden and to consequent undertreatment [35, 36]. Therapy with continuous positive airway pressure (CPAP) represents the gold standard for the treatment of OSA [37, 38]. When adherence is optimal, CPAP has been demonstrated to reduce symptoms, the possible sequelae of the disease and to improve self-reported health status [39–42]. As OSA and its related syndrome have been demonstrated to influence the onset of several health conditions and other non-clinical consequences, their low diagnosis and treatment rates are likely to result in increased clinical and economic burden. Several studies have estimated that OSA is associated with substantial economic costs, especially if untreated [43–45]. However, to date there is not available evidence for Italy. The present study aims at assessing the societal economic burden associated with OSA in the adult population in Italy by performing a cost-of-illness (COI) analysis based on a systematic literature review and population attributable fraction methodology. In particular, we aimed at estimating the proportion of the cost-of-illness of conditions for which OSA is a proven risk factor that can be attributed to OSA itself. As in the literature there is not always a clear and consistent distinction between OSA and OSAS, we focused our analyses on the broadest definition of the disease (i.e., OSA). Providing reliable estimates of the economic impact of OSA at a societal level may increase awareness of the disease burden and help to guide evidence-based policies and prioritisation for healthcare in Italy, ultimately ensuring appropriate diagnostic and therapeutic pathways for patients.

## Materials and methods

A retrospective, prevalence-based COI study with a societal perspective and a one-year time-horizon was conducted to assess the economic burden of OSA and its syndrome for the Italian adult population.

### Estimation of OSA prevalence

A scoping review of the literature, both grey and peer-reviewed, was performed in order to establish the prevalence of OSA in Italy among adult population (aged 15–74 years). Findings from the literature were discussed with clinical experts in order to assess their reliability and generalizability to the Italian context. Moreover, on the basis of the data provided by the Italian association of apneic patients (Associazione Apnoici Italiani Onlus) and expert opinion, we estimated the number of patients currently diagnosed and treated in Italy, in order to assess the extent of underdiagnosis and undertreatment in the national context.

### Identification of conditions associated with OSA

A systematic literature review was carried out in order to identify the clinical and non-clinical conditions that have been demonstrated to be influenced by OSA in the adult population. The PRISMA guideline was used in developing this review [46]. The search was performed on PubMed according to a detailed search strategy and limited to studies providing the highest level of evidence, namely systematic reviews and meta-analyses (S1 Table). We decided to perform a "systematic review of systematic reviews" [47] since this approach is recommended when the literature is extremely heterogeneous in terms of methods, definitions and results, as in the case of OSA. The search was carried out on November 19th, 2018, and was updated on May 13th, 2021. The screening and selection of titles and abstracts first and full-texts later were conducted by two researchers in parallel on the basis of pre-defined exclusion criteria (S2 Table). Disagreements between reviewers on study inclusion were solved by consensus or by the decision of a third independent reviewer. The references and citations of the full-texts included were reviewed for additional articles according to a snowballing approach in order to ensure exhaustiveness of the review. Ultimately, studies were included if they provided quantitative evidence on the influence of OSA and its syndrome on other clinical and non-clinical conditions in the adult population. From the selected studies, the following data were extracted according to a predefined template: authors and year of publication; condition investigated (e.g., diabetes); OSA's severity (i.e., mild, moderate, severe, overall); reference population (i.e., men, women, all); association measure (i.e., relative risk, hazard ratio, odds ratio); magnitude of association (mean value and 95% confidence intervals—CIs); statistical significance of the association (i.e., p-value). Data on magnitude of association (mean values) were used to calculate the proportion of conditions' epidemiological burden influenced by OSA (see next section) and 95% CIs were used for sensitivity analysis. The exclusion criteria adopted and the results of the systematic review were discussed within a research board composed of health economists and clinicians specialized in different disciplines strongly related to OSA (i.e., neurology, endocrinology, internal medicine with cardiology specialization, gastroenterology). The aim of this phase was to ensure that the conditions retrieved through the review were significant from a clinical standpoint. Ultimately, only conditions for which a statistically significant association with OSA was found (i.e., p-value<0.05) and judged clinically meaningful by experts were included in the next steps of the analysis.

## Estimation of conditions' burden influenced by OSA: Population attributable fraction

An extensive review of both published and grey literature was performed to collect Italian epidemiological data of conditions included. In case we could not retrieve an epidemiological study for Italy, we searched for studies referred to other countries. In case more than one study was available for the same condition, we considered the one providing most up-to-date estimates. When prevalence rates were reported, the total number of prevalent cases was derived using data on Italian population by age and sex provided by the Italian National Institute of Statistics for 2018 (Istat) [48].

In order to estimate the proportion of conditions' epidemiological burden influenced by OSA and its syndrome, a Population Attributable Fraction (PAF) methodology was applied. The PAF can be defined as the proportional reduction in average disease risk that would be achieved by eliminating the exposure to a risk factor [49, 50]. The PAF allows to estimate the amount of disease burden caused by a certain risk factor. In the literature, there are different approaches to estimate PAF. In the present analysis, we chose the approach that was deemed more suitable according to the association data available (i.e., relative risk, odds ratio or hazard ratio). In particular, we used the formula proposed by Levin (1953) [51] when the measure of association provided was relative risk (RR):

$$PAF = \frac{p_{(E)}(RR - 1)}{p_{(E)}(RR - 1) + 1}$$

where $p_{(E)}$ is the prevalence of OSA; RR is relative risk.

It is important to highlight that Levin's approach could lead to overestimation of PAF when the measure of association provided is adjusted RR [49]. However, studies included did not provide sufficient data to use alternative approaches, suitable in the presence of confounding, therefore we used Levin's formula for both unadjusted and adjusted RR.

When the measure of association was hazard ratio (HR), a variant of the Levin's formula was considered [52]:

$$PAF = \frac{p_{(E)}(HR(t) - 1)}{p_{(E)}(HR(t) - 1) + 1}$$

where $p_{(E)}$ is the prevalence of OSA; HR(t) is hazard ratio at time t.

Finally, when the measure of association provided was odds ratio (OR), PAF was calculated according to the method based on Eide and Heunch (2001) [53] and used in a study by Hillman and colleagues (2018) [43]. By solving simultaneously the following two equations for $p_{(D|E)}$ and $p_{(D|\sim E)}$

$$p_{(D|E)} * p_{(E)} + p_{(D|\sim E)} * p_{(\sim E)} = p_{(D)}$$

$$\left(\frac{p_{(D|E)}}{1 - p_{(D|E)}}\right) / \left(\frac{p_{(D|\sim E)}}{1 - p_{(D|\sim E)}}\right) = OR$$

the formula for PAF calculation is obtained

$$PAF = \frac{\left(p_{(D|E)} - p_{(D|\sim E)}\right) * p_{(E)}}{p_{(D)}}$$

where $p_{(D|E)}$ is the probability of having the particular condition given that an individual has

OSA; $p_{(D|\sim E)}$ is the probability of having the particular condition given that an individual does not have OSA; $p_{(E)}$ is the probability of having OSA (i.e., OSA prevalence); $p_{(\sim E)}$ is the probability of not having OSA; $p_{(D)}$ is the probability of having the particular condition; OR is the odds ratio for that condition for individuals with OSA.

The number of prevalent cases for each condition influenced by OSA and its syndrome was obtained by multiplying conditions' prevalence by the relative PAF.

## Cost analysis: Assessment of conditions' costs and estimation of OSA economic burden

An extensive literature review on scientific databases (e.g., ScienceDirect, Scopus) and on Google was carried out in order to retrieve Italian cost data for the conditions included in the systematic review. A top-down approach was used for cost estimation. As a societal perspective was adopted, all cost categories (i.e., direct healthcare, direct non-healthcare and indirect costs) were included, when available. In case we could not retrieve a cost study for Italy, we included cost studies referred to other countries, adjusting for inflation and purchasing power differences. To estimate indirect costs due to all-cause mortality associated with OSA, the friction method was used. This method assumes that for long term absences, as in the case of premature death, an individual's work can be replaced by the market, therefore the loss in production is limited to a period in which the market adapts to the changed situation, called friction period [54]. This approach is more conservative than the human capital method, which considers earnings lost over a lifetime [55]. In the present analysis, we considered only productivity losses of employed people, excluding indirect costs of people out of the labour market. Productivity costs were estimated for different age groups to account for differences in wages, considering age and gender-specific yearly paid production value [56] and employment rates [48]. People aged 65–74 were assumed to be retired, therefore their employment rate was set equal to 0. To the best of our knowledge, there are not published data on the friction period for Italy, therefore we used a plausible value of 75 days, in line with the estimates used in another European country (Spain) [57]. Moreover, for employed people, we assumed that the friction period was the same across all age groups. The mean cost per patient/year was calculated for each condition and multiplied for the number of prevalent cases influenced by OSA and its syndrome, as obtained by applying PAF methodology.

As OSA is associated with decreased quality of life (QoL) [18–20], we estimated its burden also in terms of quality-adjusted life years (QALYs) value lost due to OSA undertreatment. QALYs for a single patient can be obtained by multiplying the health utility values for the years lived [58]. Health utilities represent individuals' preferences for different health states and can take on values from 0 (death) to 1 (perfect health) [58]. Since our time perspective is one year, in the present case the QALY for a single patient coincides with the health utility value. In order to express the QALYs lost due to undertreatment in monetary terms, we evaluated them using a willingness-to-pay (WTP) threshold, which represents a measure of the amount of money a society is willing to invest in order to improve health (in this case, to obtain one additional QALY). Health utility and WTP values were retrieved from a review of the literature. In particular, the WTP threshold was sourced from an empirical work carried out by Woods and colleagues [59]. We adopted a conservative approach and considered the lower WTP value reported by the authors (i.e., $16,712, corresponding to €14,860 in 2018). In line with expert opinion, we assumed that only moderate-severe OSA has a substantial impact on patients' QoL, therefore the present analysis was focused on this patient subpopulation. QALYs value lost was computed according to the following formulas, for alive and dead

moderate-severe patients respectively:

$$QALYs\ value\ lost_{alive} = [(utility_{treated} - utility_{untreated}) * alive\ untreated\ patients] * WTP$$

$$QALYs\ value\ lost_{dead}$$
$$= \left[ \left( utility_{treated} - \left( \frac{utility_{untreated}}{2} + \frac{utility_{dead}}{2} \right) \right) * dead\ untreated\ patients \right] * WTP$$

The total economic burden of OSA for the society was estimated by summing the costs of each condition influenced by the disease, the costs due to OSA's diagnosis and treatment and the QALYs value lost.

All costs were adjusted for inflation to 2018 (most updated data when analysis was carried out) in their national currency using gross domestic product deflators retrieved from the Organisation for Economic Co-operation and Development (OECD) database [60]. Finally, all costs were adjusted for purchasing power differences using OECD Purchasing Power Parities (PPPs) for GDP for 2018 [60]. PPPs serve both as currency convertors and as spatial price deflators: they convert different currencies to a common currency and, in the process of conversion, equalise their purchasing power by eliminating the differences in price levels between countries. This methodology can ensure better comparability between different currencies. As a one-year time-horizon was considered, no discounting on costs was applied.

## Sensitivity analysis

A one-way deterministic sensitivity analysis was performed in order to account for uncertainty and test robustness of results regarding conditions' burden influenced by OSA. In particular, all key parameters of the analysis were tested and varied according to 95% confidence intervals (95% CIs) or plausible ranges of variation: OSA prevalence (±10%), conditions' prevalence (±10%), magnitude of association (95% CIs) and conditions' costs (±10%). Each variable was tested at the upper and lower limits of its respective interval. Results were graphically reported through a tornado diagram.

## Results

### OSA prevalence in Italy

The review of the literature revealed that epidemiological studies on OSA in Italy are scant [61–63], and the estimates provided were either outdated or focused on a local context, therefore hardly generalizable. Moreover, as OSA is a highly undiagnosed condition, epidemiological studies conducted on sample of individuals with suspected OSA are likely to be biased and capture only diagnosed prevalence, thus underestimating the real one. On the basis of clinical expert opinion, two European-based and one literature-based studies were included for OSA's prevalence estimate in the Italian population aged 15–74 years. In particular, prevalence data provided by one of the European studies identified (HypnoLaus [31]), a population-based study, were considered both representative of OSA epidemiology in Italy, reflecting the prevalence ratio of 2:1 among men and women observed in the Italian adult population [63], and reliable (i.e., reflecting the actual prevalence), as the sample of individuals tested with polysomnography was randomly drawn from the general population, considering all individuals regardless any OSA suspicions. The literature-based study identified [64], which used publicly available data and expert opinion to estimate the global prevalence of OSA, provided lower prevalence data for Italy than those obtained from the HypnoLaus. Therefore, in order to take into account the full range of variation of prevalence estimates, both studies were included in

our analyses. The other European-based study by Hedner and colleagues [65] was used to stratify patients according to OSA severity as measured by AHI. Finally, data on the resident population in Italy in 2018 were sourced from Istat [48] and used to derive the number of prevalent cases. Additional information on prevalence estimates are reported in S1 File.

Results suggest that OSA prevalence in Italy is substantial, with moderate-severe condition (AHI≥15) affecting between 9% and 27% of the population aged 15–74 years (Table 1). On the basis of the data collected by the Italian association of apneic patients (Associazione Apnoici Italiani Onlus), in Italy patients currently treated with continuous and automatic positive air-way pressure (the standard of care), are approximately 230,000, representing around 2% (model 1) and 6% (model 2) of the estimated prevalence of moderate-severe OSA. According to expert opinion, the patients currently diagnosed are approximately twofold than those treated (around 4% (model 1) and 12% (model 2) of moderate-severe OSA patients), but still far from the actual prevalence rates. These data suggest a substantial gap in both OSA diagnosis and treatment.

## Conditions influenced by OSA

Of the 702 studies retrieved, 86 were included for full-text reading (Fig 1), as they did not meet any of the exclusion criteria previously established (S2 Table). If more than one study was available for the same condition, only the most recent and comprehensive one was included. However, if studies on the same condition reported discordant results on the association with OSA, they were all included in the analysis. On the basis of full-texts analysis and reference screening, 23 meta-analyses were selected [66–88] (Fig 1). Unfortunately, for some conditions judged relevant by clinicians involved in the research board (e.g., arrhythmias, psoriasis), screened studies either did not provide sufficient quantitative data on the association or provided an association measure that could not be used for PAF calculation (e.g., Cohen's d, rate ratio), therefore they were excluded from the analysis. For each condition included, data were extracted according to a predefined template presented in the Methods section (Table 2). Four conditions included in data extraction but either judged not clinically meaningful by experts or for which there is a lack of epidemiological/cost evidence (i.e., spontaneous cerebrospinal fluid leak, floppy eyelids syndrome, nonarteritic anterior ischemic optic neuropathy, pulmonary edema during pregnancy) were ultimately excluded from COI analysis, as well as conditions with a non-statistically significant association (Table 2). Overall, 26 clinical and non-clinical conditions from 18 meta-analyses were considered for the estimation of OSA's economic burden (Fig 1, Table 2). The PRISMA checklist [46] is provided in the S2 File.

## Conditions' burden associated with OSA

For the conditions included, epidemiological data were retrieved from different sources [48, 89–110] and reported in S3 File. Conditions' prevalence data (or incidence when appropriate) were used, together with data on magnitude of association (Table 2) and prevalence of OSA (Table 1), to estimate the proportion of burden associated with OSA through PAF methodology [43, 51, 52]. The PAF for car and work accidents was derived considering only OSA population with excessive daytime sleepiness (EDS), estimated at 19% [35]. Moreover, for the conditions providing only estimates for overall OSA, a conservative approach was adopted and PAF was calculated considering moderate-severe subpopulation as, according to expert opinion, these patients are more likely to develop comorbidities than mild ones. Table 3 provides the results for PAF (i.e., the proportion of each condition influenced by the presence of OSA and its syndrome) and the total number of cases/year for each condition, calculated using

**Table 1. Prevalence of OSA for the general adult population in Italy (aged 15–74).**

| | Estimates from the population-based study (model 1) | | | Estimates from the literature-based study (model 2) | | |
|---|---|---|---|---|---|---|
| | Female | Male | Total | Female | Male | Total |
| *Rates* | | | | | | |
| Mild (5≤AHI<15) | 29.2% | 24.8% | 27.1% | 7.2% | 5.2% | 6.2% |
| Moderate-severe (AHI≥15) | 18.3% | 36.2% | 27.1% | 5.9% | 11.7% | 8.8% |
| Moderate (15≤AHI<30) | 9.8% | 14.5% | 12.1% | 3.2% | 4.7% | 3.9% |
| Severe (AHI≥30) | 8.5% | 21.7% | 15.0% | 2.8% | 7.0% | 4.9% |
| Overall (AHI≥5) | 47.5% | 61.0% | 54.2% | 13.1% | 16.9% | 15.0% |
| *Absolute values* | | | | | | |
| Mild (5≤AHI<15) | 6,703,067 | 5,582,051 | 12,285,118 | 1,657,025 | 1,163,534 | 2,820,559 |
| Moderate-severe (AHI≥15) | 4,193,897 | 8,135,717 | 12,329,614 | 1,354,459 | 2,627,507 | 3,981,966 |
| Moderate (15≤AHI<30) | 2,236,745 | 3,260,161 | 5,496,906 | 722,378 | 1,052,900 | 1,775,278 |
| Severe (AHI≥30) | 1,957,152 | 4,875,556 | 6,832,708 | 632,081 | 1,574,607 | 2,206,688 |
| Overall (AHI≥5) | 10,896,964 | 13,717,768 | 24,614,732 | 3,011,484 | 3,791,041 | 6,802,526 |

Source. Our elaboration from Heinzer et al (2015) [31], Benjafield et al (2019) [64], Hedner et al (2011) [65] and Istat data [48].

OSA prevalence data from either the population-based (model 1) or the literature-based study (model 2), and stratified by OSA severity.

## OSA economic burden

For all-cause and cardiovascular premature mortality, indirect costs were estimated using the friction cost method, for both model 1 and model 2 (S4 File). Cost data of all other conditions were retrieved from the literature [107, 111–129] and expressed in 2018 Euros standardized for inflation and PPP (S5 File). In order to avoid double counting, only productivity losses due to morbidity were considered for these conditions when the original study reported separate estimates for costs due morbidity and mortality, as indirect costs due to mortality were computed separately. Unfortunately, for some conditions, it was not possible to retrieve an Italian cost study. Moreover, the majority of studies did not report estimates for all cost categories, leading to a possible underestimation of the overall economic burden. Through the multiplication of the cost per patient/year by the number of prevalent (or incident) cases due to OSA, we estimated the total economic burden influenced by OSA in Italy in one year. Mean estimates are provided in Table 4. Results suggest that the economic burden due to conditions influenced by OSA in Italy is substantial and ranges from €10.7 billion (model 2) to €32.0 billion (model 1) per year, corresponding to €177 and €530 per Italian resident respectively. The main driver of economic burden are direct healthcare costs, which account for 60% and 57% of total cost according to model 1 and model 2 respectively, followed by indirect costs (37% and 39%) and direct non-healthcare costs (both 4%). Considering the health expenditure per capita in Italy in 2018 (€3,429) [130], the direct healthcare costs per resident generated by conditions influenced by OSA represent between the 3% (model 2) and 9% (model 1) of national health expenditure.

The yearly per patient cost of OSA diagnosis and treatment in Italy amounts to approximately €381 and €256, respectively (see S6 File for all details on data sources and calculation). According to the Italian association of apneic patients (Associazione Apnoici Italiani Onlus) and expert opinion, in Italy there are approximately 460,000 patients diagnosed and 230,000 treated. Therefore, the total yearly cost of OSA diagnosis and treatment amounts to €175,347,041 and €58,880,000 respectively, with an overall cost of €234,227,041.

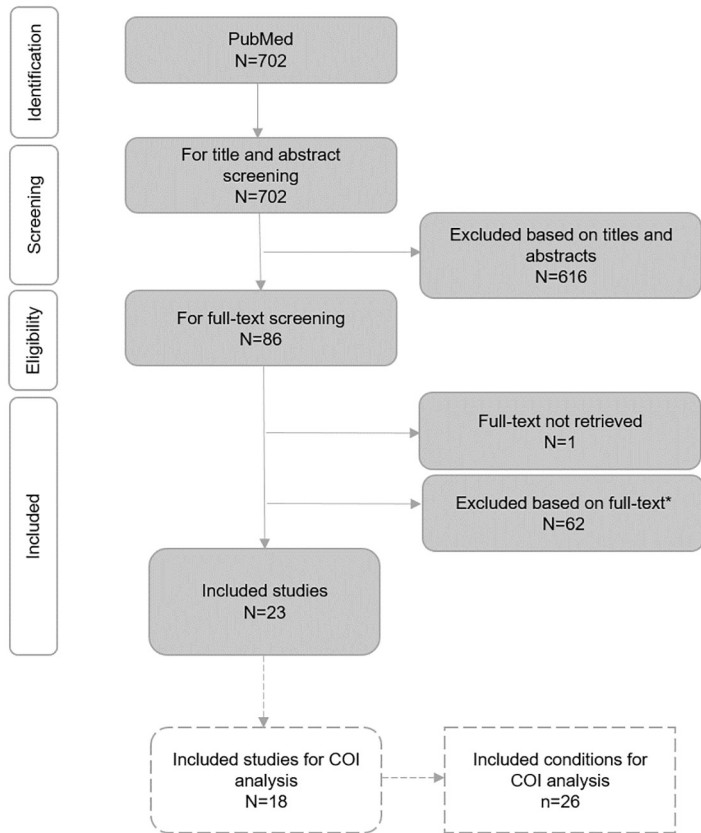

**Fig 1. Systematic literature review—Screening process (PRISMA flow diagram).** *Reasons for full-text exclusion: 1) OSA exclusively investigated as a consequence of other conditions; 2) no quantitative data provided on the association of OSA with other conditions; 3) focus on parameters that could eventually identify a clinical condition, but not on the clinical condition itself; 4) lack of a control group of non-OSA patients; 5) unclear data on the direction of the association between OSA and the condition investigated; 6) focus on OSA together with other sleep disorders; 7) measure of association that could not be used for PAF calculation; 8) association between increase in AHI and the condition investigated.

As regards estimation of QALYs value lost due to undertreatment, health utility values were derived from a study by Català and colleagues [131], who found a significant difference in utility values between treated (utility = 0.84) and untreated patients (utility = 0.79). By using these values in the formulas presented in the methods section, we estimated a QALYs value lost ranging from €2.8 billion (model 2) to €9.0 billion (model 1) (see S7 File for additional details on calculation).

As summarized by Table 5, in Italy the total economic burden of OSA ranges from around € 13.8 billion/year (model 2) to €41.3 billion/year (model 2).

## Sensitivity analysis

All key parameters used to estimate conditions' burden influenced by OSA were tested in one-way deterministic sensitivity analysis. The majority of them, however, did not significantly influence the results obtained in the base-case analysis. The tornado plot shows only those variables whose variation caused at least 1% increase or decrease of base-case result. For both model 1 (Fig 2) and model 2 (Fig 3), the five parameters with the highest impact on conditions'

**Table 2. Results of systematic literature review: Association between OSA and other conditions.**

| Condition | OSA severity | Association measure | Magnitude | 95% CIs | p-value | Included in COI analysis | Source |
|---|---|---|---|---|---|---|---|
| All-cause mortality | Mild | RR | 1.26 | 0.77–2.07 | NS | No | Xie et al (2017) [66] |
| | Moderate | RR | 1.04 | 0.60–1.79 | NS | No | |
| | Severe | RR | 1.54 | 1.21–1.97 | <0.001 | Yes | |
| Cancer mortality | Mild | HR | 0.79 | 0.46–1.34 | NS | No | Zhang et al (2017) [67] |
| | Moderate | HR | 1.92 | 0.63–5.88 | NS | No | |
| | Severe | HR | 2.09 | 0.45–9.81 | NS | No | |
| | Overall | HR | 1.38 | 0.79–2.41 | NS | No | |
| Cardiovascular mortality | Mild | RR | 1.80 | 0.68–4.76 | NS | No | Xie et al (2017) [66] |
| | Moderate | RR | 1.11 | 0.53–2.35 | NS | No | |
| | Severe | RR | 2.96 | 1.45–6.01 | 0.003 | Yes | |
| Cancer | Mild | HR | 0.91 | 0.74–1.13 | NS | No | Zhang et al (2017) [67] |
| | Moderate | HR | 1.07 | 0.86–1.33 | NS | No | |
| | Severe | HR | 1.03 | 0.85–1.26 | NS | No | |
| | Overall | HR | 1.04 | 0.92–1.16 | NS | No | |
| | Overall | RR | 1.40 | 1.01–1.95 | 0.04 | Yes | Palamaner Subash Shantha et al (2015) [68] |
| Diabetic retinopathy | Overall | OR | 2.01 | 1.49–2.72 | <0.05 | Yes | Zhu et al (2017) [69] |
| Diabetic kidney disease | Overall | OR | 1.59 | 1.16–2.18 | <0.05 | Yes | Leong et al (2016) [70] |
| Type 2 diabetes mellitus | Mild | RR | 1.22 | 0.91–1.63 | NS | No | Wang et al (2013) [71] |
| | Moderate-severe | RR | 1.63 | 1.09–2.45 | 0.018 | Yes | |
| Metabolic syndrome | Mild | OR | 2.39 | 1.65–3.46 | <0.05 | Yes | Xu et al (2015) [72] |
| | Moderate-severe | OR | 3.42 | 2.28–5.13 | <0.05 | Yes | |
| Depression | Overall | OR[†] | 1.12 | 0.78–1.47 | NS | No | Edwards et al (2020) [73] |
| | | RR[‡] | 2.18 | 1.47–2.88 | 0.005 | Yes | |
| Erectile dysfunction | Overall (men) | RR | 1.82 | 1.12–2.97 | <0.05 | Yes | Liu et al (2015) [74] |
| Female sexual dysfunction | Overall (women) | RR | 2.00 | 1.29–3.08 | <0.05 | Yes | Liu et al (2015) [74] |
| Parkinson's disease | Overall | HR | 1.59 | 1.36–1.85 | <0.001 | Yes | Sun et al (2020) [75] |
| Stroke | Mild | RR | 1.29 | 0.69–2.41 | NS | No | Xie et al (2017) [66] |
| | Moderate | RR | 1.35 | 0.82–2.23 | NS | No | |
| | Severe | RR | 2.15 | 1.42–3.24 | <0.001 | Yes | |
| Spontaneous cerebrospinal fluid leak | Overall | OR | 3.43 | 1.55–7.59 | 0.002 | No | Bakhsheshian et al (2015) [76] |
| Floppy eyelids syndrome | Overall | OR | 4.70 | 2.98–7.41 | <0.001 | No | Huon et al (2016) [77] |
| Glaucoma | Overall | OR | 1.24 | 1.20–1.28 | <0.001 | Yes | Huon et al (2016) [77] |
| Nonarteritic anterior ischemic optic neuropathy | Overall | OR | 6.18 | 2.00–19.11 | 0.002 | No | Wu et al (2016) [78] |
| Resistant hypertension | Overall | OR | 2.84 | 1.70–3.98 | <0.05 | Yes | Hou et al (2018) [79] |
| Essential hypertension | Mild | OR | 1.18 | 1.09–1.27 | <0.05 | Yes | Hou et al (2018) [79] |
| | Moderate | OR | 1.32 | 1.20–1.43 | <0.05 | Yes | |
| | Severe | OR | 1.56 | 1.29–1.84 | <0.05 | Yes | |
| Ischemic heart disease | Mild | RR | 1.25 | 0.95–1.66 | NS | No | Xie et al (2017) [66] |
| | Moderate | RR | 1.38 | 1.04–1.83 | 0.026 | Yes | |
| | Severe | RR | 1.63 | 1.18–2.26 | 0.003 | Yes | |
| Heart failure | Mild | RR | 1.02 | 0.78–1.34 | NS | No | Xie et al (2017) [66] |
| | Moderate | RR | 1.07 | 0.74–1.54 | NS | No | |
| | Severe | RR | 1.44 | 0.94–2.21 | NS | No | |

*(Continued)*

**Table 2.** (Continued)

| Condition | OSA severity | Association measure | Magnitude | 95% CIs | p-value | Included in COI analysis | Source |
|---|---|---|---|---|---|---|---|
| Aortic dissection | Mild | OR | 1.60 | 1.01–2.53 | 0.04 | Yes | Zhou et al (2018) [80] |
| | Moderate-severe | OR | 4.43 | 2.59–7.59 | <0.001 | Yes | |
| Allergic rhinitis | Overall | OR | 1.73 | 0.94–3.20 | NS | No | Cao et al (2018) [81] |
| Non-alcoholic fatty liver disease | Overall | OR | 2.34 | 1.71–3.18 | <0.001 | Yes | Musso et al (2013) [82] |
| Gastroesophageal reflux disease | Overall | OR | 1.57 | 1.07–2.08 | <0.05 | Yes | Wu et al (2018) [83] |
| Gout | Overall | HR | 1.25 | 0.91–1.70 | NS | No | Shi et al (2019) [84] |
| Pre-eclampsia | Overall (women) | OR | 2.35 | 2.15–2.58 | <0.001 | Yes | Liu et al (2019) [85] |
| Gestational hypertension | Overall (women) | OR | 1.97 | 1.51–2.56 | <0.001 | Yes | Liu et al (2019) [85] |
| Gestational diabetes | Overall (women) | RR | 1.40 | 0.62–3.19 | NS | No | Xu et al (2014) [86] |
| | Overall (women) | OR | 1.55 | 1.26–1.90 | <0.001 | Yes | Liu et al (2019) [85] |
| Preterm delivery | Overall (women) | OR | 1.62 | 1.29–2.02 | <0.001 | Yes | Liu et al (2019) [85] |
| Cesarean delivery | Overall (women) | OR | 1.42 | 1.12–1.79 | <0.001 | Yes | Liu et al (2019) [85] |
| Pulmonary edema | Overall (women)* | OR | 6.35 | 4.25–9.50 | <0.001 | No | Liu et al (2019) [85] |
| Car accidents | Overall | OR | 2.43 | 1.21–4.89 | 0.013 | Yes | Tregear et al (2009) [87] |
| Work accidents | Overall | OR | 1.78 | 1.03–3.07 | <0.001 | Yes | Garbarino et al (2016) [88] |

Note.

[†]Estimates obtained from a meta-analysis of cross-sectional studies.

[‡]Estimates were obtained from a meta-analysis of longitudinal studies.

*The focus was only on pregnant women.

burden influenced by OSA were the magnitude of association with OSA of metabolic syndrome, non-alcoholic fatty liver disease, type 2 diabetes and cancer, and OSA prevalence.

## Discussion

This study aimed at providing reliable estimates of the extent of OSA consequences in Italy and assessing the societal economic burden associated with OSA in the adult population by performing a COI analysis. Several studies demonstrated that OSA is a severely underdiagnosed condition worldwide [1, 132, 133]. We estimated a prevalence of moderate-severe OSA in Italy ranging from approximately 4 to 12 million patients (9% to 27% of the adult population). However, the number of diagnosed and treated patients is substantially lower (around 460,000 and 230,000 patients respectively), suggesting a huge gap in both OSA diagnosis and treatment. The reasons underlying poor diagnosis of OSA are several, and start from lack of awareness [134, 135], both among healthcare professionals and general population, to limited routine screening and diagnostic sleep centres [136]. Even when a diagnosis occurred, evidence shows that acceptance and adherence to treatment with CPAP—despite its technological advances—is generally low, ranging from 30 to 60% [137, 138].

The systematic literature review revealed that several clinical and non-clinical conditions were found to be significantly influenced by OSA and its syndrome. Through the PAF approach, we attributed a portion of each conditions' burden to OSA, which allowed us to

**Table 3. Conditions' burden influenced by OSA: PAF and number of cases/year among general adult population in Italy (aged 15–74 years).**

| Condition | OSA severity | Model 1 | | Model 2 | | Source of epidemiological data |
|---|---|---|---|---|---|---|
| | | PAF | Number of cases/year | PAF | Number of cases/year | |
| All-cause mortality | Severe | 7.5% | 11,129 | 2.6% | 3,797 | Istat [48] |
| Cardiovascular mortality | Severe | 22.7% | 7,377 | 8.7% | 2,823 | Istat [48] |
| Cancer | Overall* | 9.7% | 184,224 | 3.4% | 64,038 | AIOM-AIRTUM (2018) [89] |
| Diabetic retinopathy | Overall* | 20.7% | 246,930 | 7.8% | 92,650 | AMD et al (2015) [90] |
| Diabetic kidney disease | Overall* | 13.5% | 92,915 | 4.8% | 33,131 | AMD-SID (2018) [91] |
| | | | | | | IDF (2017) [92] |
| Type 2 diabetes | Moderate-severe | 14.5% | 450,426 | 5.2% | 162,188 | IDF (2017) [92] |
| Metabolic syndrome | Mild | 16.4% | 2,454,641 | 3.9% | 587,706 | Tocci et al (2015) [93] |
| | Moderate-severe | 23.2% | 3,470,981 | 7.8% | 1,171,728 | |
| Depression□ | Overall* | 24.2% | 175,121 | 9.4% | 67,954 | Istat (2018) [94] |
| Erectile dysfunction | Overall (men)* | 22.8% | 511,257 | 8.7% | 196,081 | Nicolosi et al (2003) [95] |
| Female sexual dysfunction | Overall (women)* | 15.3% | 1,014,992 | 5.6% | 371,226 | Graziottin (2007) [96] |
| Parkinson's disease | Overall* | 13.7% | 7,726 | 4.9% | 2,766 | Riccò et al (2020) [97] |
| Stroke† | Severe | 14.7% | 10,757 | 5.3% | 3,869 | Stevens et al (2017) [98] |
| Glaucoma | Overall* | 6.0% | 48,430 | 2.0% | 16,373 | Tham et al (2014) [99] |
| Resistant hypertension | Overall* | 32.5% | 235,129 | 13.4% | 97,135 | Giampaoli et al (2015) [100] |
| | | | | | | Dovellini (2000) [101] |
| Essential hypertension | Mild | 3.2% | 442,561 | 0.8% | 103,155 | Giampaoli et al (2015) [100] |
| | Moderate | 2.4% | 327,235 | 0.8% | 107,445 | |
| | Severe | 4.9% | 673,131 | 1.6% | 221,359 | Dovellini (2000) [101] |
| Ischemic heart disease | Moderate | 4.4% | 99,296 | 1.5% | 33,325 | Giampaoli et al (2015) [100] |
| | Severe | 8.6% | 196,584 | 3.0% | 67,625 | |
| Aortic dissection† | Mild | 13.9% | 224 | 3.6% | 58 | Pacini et al (2013) [102] |
| | Moderate-severe | 48.1% | 774 | 23.1% | 372 | |
| Non-alcoholic fatty liver disease | Overall* | 19.9% | 1,844,121 | 7.0% | 653,912 | Younossi et al (2016) [103] |
| Gastroesophageal reflux disease | Overall* | 11.0% | 536,097 | 3.8% | 185,754 | Darbà et al (2011) [104] |
| Pre-eclampsia | Overall (women)* | 19.5% | 1,790 | 7.4% | 676 | Fox et al (2017) [105] |
| Gestational hypertension | Overall (women)* | 14.9% | 2,041 | 5.4% | 744 | FIGO (2016) [106] |
| Gestational diabetes | Overall (women)* | 9.0% | 4,486 | 3.1% | 1,567 | Meregaglia et al (2018) [107] |
| Preterm delivery | Overall (women)* | 10.0% | 2,801 | 3.5% | 986 | Merinopoulou et al (2018) [108] |
| Cesarean delivery | Overall (women)* | 7.0% | 11,525 | 2.4% | 3,967 | OECD [109] |
| Car accidents ‡ | Overall* | 5.3% | 11,420 | 2.1% | 92,650 | Istat-Aci (2017) [110] |
| Work accidents ‡ | Overall* | 3.3% | 845 | 1.2% | 33,131 | Istat-Aci (2017) [110] |

Note. Model 1: Statistics calculated using OSA prevalence data derived from the population-based study. Model 2: Statistics calculated using OSA prevalence data derived from the literature-based study.

□Estimates are referred to major depression.

†Incidence data were considered.

‡Only OSA population with excessive daytime sleepiness was considered for PAF calculation.

*Only conservative estimates (i.e., referred to moderate-severe subpopulation) were provided.

estimate the economic impact associated with the sleep disorder. Results of COI revealed that the economic burden for the society due to conditions associated with OSA in Italy is very high, ranging from €10.7 to €32.0 billion per year, with the main cost driver represented by direct healthcare costs. Moreover, the QALYs value lost due to OSA undertreatment is substantial, and contributes at increasing OSA's yearly economic burden by €2.8 to €9.0 billion.

**Table 4. Annual economic burden of conditions influenced by OSA in Italy.**

| Condition | Model 1 | | | | Model 2 | | | |
|---|---|---|---|---|---|---|---|---|
| | Direct healthcare cost | Direct non-healthcare cost | Productivity losses cost* | Total cost | Direct healthcare cost | Direct non-healthcare cost | Productivity losses cost* | Total cost |
| Mortality † | | | € 17,468,314 | € 17,468,314 | | | € 5,960,283 | € 5,960,283 |
| Cancer | € 1,053,335,086 | € 843,866,787 | € 21,976,498 | € 1,919,178,370 | € 366,149,501 | € 293,336,287 | € 7,639,244 | € 667,125,033 |
| Diabetic retinopathy | € 75,903,881 | € 59,787,476 | € 142,872,419 | € 278,563,776 | € 28,479,821 | € 22,432,800 | € 53,607,020 | € 104,519,641 |
| Diabetic kidney disease | € 74,076,551 | | | € 74,076,551 | € 26,413,493 | | | € 26,413,493 |
| Type 2 diabetes | € 1,741,141,727 | | € 1,960,219,449 | € 3,701,361,176 | € 626,945,056 | | € 705,829,901 | € 1,332,774,957 |
| Metabolic syndrome | € 11,260,422,980 | | € 531,818,651 | € 11,792,241,631 | € 3,343,442,091 | | € 157,907,466 | € 3,501,349,558 |
| Depression□ | € 152,472,494 | € 86,853,842 | € 340,910,896 | € 580,237,232 | € 59,165,781 | € 33,702,967 | € 132,287,858 | € 225,156,606 |
| Erectile dysfunction | € 208,151,669 | | | € 208,151,669 | € 79,831,780 | | | € 79,831,780 |
| Female sexual dysfunction | € 772,808,563 | | | € 772,808,563 | € 282,649,080 | | | € 282,649,080 |
| Parkinson's disease | € 47,486,623 | € 37,281,979 | € 9,360,588 | € 94,129,190 | € 16,997,509 | € 13,344,827 | € 3,350,558 | € 33,692,894 |
| Stroke | € 144,697,413 | € 91,324,306 | € 9,755,712 | € 245,777,431 | € 52,047,358 | € 32,849,162 | € 3,509,109 | € 88,405,629 |
| Glaucoma | € 47,690,291 | | | € 47,690,291 | € 16,122,559 | | | € 16,122,559 |
| Resistant hypertension | € 56,172,997 | | | € 56,172,997 | € 23,205,874 | | | € 23,205,874 |
| Essential hypertension | € 344,718,771 | | | € 344,718,771 | € 103,196,342 | | | € 103,196,342 |
| Ischemic heart disease | € 442,622,880 | € 103,029,886 | € 135,642,496 | € 681,295,262 | € 151,016,288 | € 35,152,252 | € 46,279,185 | € 232,447,725 |
| Aortic dissection | € 37,984,396 | | | € 37,984,396 | € 16,358,614 | | | € 16,358,614 |
| Non-alcoholic fatty liver disease | € 2,208,249,940 | | € 8,158,942,384 | € 10,367,192,324 | € 783,029,477 | | € 2,893,102,030 | € 3,676,131,507 |
| Gastroesophageal reflux disease | € 165,097,914 | | € 99,881,300 | € 264,979,215 | € 57,205,202 | | € 34,608,129 | € 91,813,331 |
| Pre-eclampsia | € 8,356,628 | | | € 8,356,628 | € 3,157,267 | | | € 3,157,267 |
| Gestational hypertension | € 24,202,545 | | | € 24,202,545 | € 8,824,400 | | | € 8,824,400 |
| Gestational diabetes | € 16,844,415 | | | € 16,844,415 | € 5,883,403 | | | € 5,883,403 |
| Preterm delivery | € 25,281,917 | | € 27,402,594 | € 52,684,511 | € 8,897,225 | | € 9,643,535 | € 18,540,759 |
| Cesarean delivery | € 28,991,021 | | € 10,865,180 | € 39,856,201 | € 9,979,392 | | € 3,740,051 | € 13,719,443 |
| Car accidents | € 106,754,952 | | € 272,184,561 | € 378,939,513 | € 42,564,671 | | € 108,523,736 | € 151,088,407 |
| Work accidents | € 7,900,413 | | € 20,143,051 | € 28,043,464 | € 2,906,425 | | € 7,410,278 | € 10,316,703 |
| **Total** | **€ 19,051,366,068** | **€ 1,222,144,276** | **€ 11,759,444,093** | **€ 32,032,954,437** | **€ 6,114,468,608** | **€ 430,818,296** | **€ 4,173,398,383** | **€ 10,718,685,287** |

Note. Model 1: Statistics calculated using OSA prevalence data derived from the population-based study. Model 2: Statistics calculated using OSA prevalence data derived from the literature-based study.

†Costs due to cardiovascular mortality were included in all-cause mortality costs in order to avoid double counting.

□Estimates are referred to major depression.

*Only productivity losses due to morbidity were included for conditions different from mortality when the original study reported separate estimates for costs due morbidity and mortality.

**Table 5. Total annual economic burden of OSA in Italy.**

|  | Model 1 | Model 2 |
|---|---|---|
| OSA diagnosis and treatment | € 234,227,041 | € 234,227,041 |
| Conditions influenced by OSA | € 32,032,954,437 | € 10,718,685,287 |
| QALYs value lost | € 9,029,365,722 | € 2,801,136,971 |
| **Total economic burden** | **€ 41,296,547,200** | **€ 13,754,049,299** |

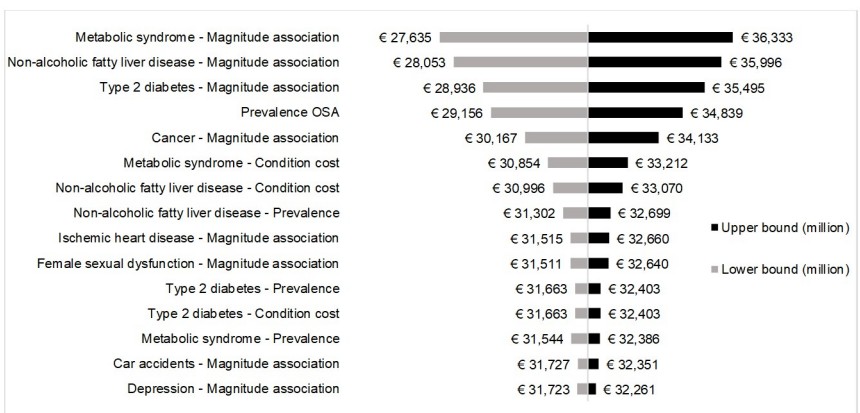

**Fig 2. Tornado plot for sensitivity analysis (model 1).**

Several studies demonstrated that appropriate diagnostic and therapeutic pathways for OSA may have a substantial impact in reducing clinical and non-clinical consequences related to the disease, whereas untreated disease may result in increased clinical and economic burden [3, 139]. In particular, therapy with CPAP was demonstrated to be effective in preventing the onset and reducing the burden of some of the associated conditions, including all-cause and cardiovascular mortality [140], stroke [141], car [142] and work accidents [143]. Overall, these results suggest that OSA's underdiagnosis and undertreatment have a detrimental effect both

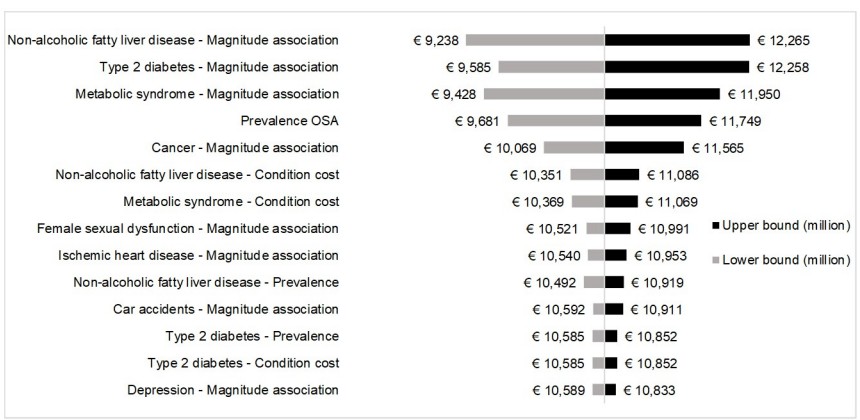

**Fig 3. Tornado plot for sensitivity analysis (model 2).**

on the onset of associated conditions and on patients' quality of life, ultimately leading to loss of value for the society. An increasing awareness towards the disease is fundamental in order to implement appropriate diagnostic and treatment pathways for OSA patients and reduce its substantial clinical and economic burden.

## Study limitations

This study has some limitations. First, the review performed may not be fully exhaustive as we did not consider individual studies but focused only on systematic reviews and meta-analyses, including the latter for COI analysis. The reason underlying this choice is that evidence on OSA's association with other conditions is very heterogeneous, therefore we opted for the "systematic review of systematic reviews" method [47], which is recommended in all cases where individual studies are heterogeneous in terms of methods, definitions and results. Moreover, we further restricted the selection on systematic reviews presenting a meta-analysis because we needed a reliable quantitative assessment of the intensity of association between OSA and other conditions. Although a vast literature exists on OSA association with other conditions, in some cases single studies either provide insufficient quantitative evidence or no quantitative evidence at all. Second, due to lack of Italian data, we had to rely on OSA epidemiological estimates derived from two different studies, a population-based study conducted in another European country and a model-based study. Estimated prevalence data, although validated by a clinical expert, should be interpreted cautiously. Further research is needed in order to provide up-to-date evidence on OSA epidemiology in Italy, which in turn might increase awareness of the extent of conditions' burden for our country. Third, few conditions' prevalence data were lacking for Italy, and we had to use estimates derived from other countries. The same happened for cost data, although we adjusted for purchasing power differences to ensure better comparability between different currencies. Moreover, it was not always possible to retrieve data on all relevant cost components, namely direct non-healthcare costs and productivity losses due to morbidity, potentially leading to an underestimation of OSA economic burden.

## Conclusions

Results of the present COI analysis suggest that the burden of OSA in Italy is substantial but subtle, because it is greatly hidden behind the cost of other conditions for which OSA is a risk factor. Moreover, underdiagnosis and low treatment rates are observed. More appropriate diagnosis rates and clinical pathways for OSA patients, in particular for moderate-severe population, are recommended.

## Supporting information

**S1 Table. Search strategy.**
(DOCX)

**S2 Table. Exclusion criteria.**
(DOCX)

**S1 File. Additional information on OSA prevalence estimation.**
(DOCX)

**S2 File. PRISMA checklist.**
(DOCX)

**S3 File. Prevalence of conditions significantly associated with OSA.**
(DOCX)

**S4 File. Indirect costs due to all-cause and cardiovascular mortality.**
(DOCX)

**S5 File. Cost of conditions.**
(DOCX)

**S6 File. Cost of OSA diagnosis and treatment.**
(DOCX)

**S7 File. Additional information on QALYs value lost calculation.**
(DOCX)

## Acknowledgments

The authors gratefully acknowledge Prof. Livio Luzi, Prof. Nicola Montano and Prof. Roberto Penagini for their participation in the research board and for their precious feedbacks to ameliorate the research. The authors thank the Italian association of apneic patients (Associazione Apnoici Italiani Onlus) for having shared their data, which contributed at enriching the analysis.

## Author Contributions

**Conceptualization:** Ludovica Borsoi, Patrizio Armeni, Gleb Donin, Francesco Costa.

**Data curation:** Ludovica Borsoi, Gleb Donin.

**Formal analysis:** Ludovica Borsoi.

**Funding acquisition:** Patrizio Armeni.

**Methodology:** Ludovica Borsoi, Patrizio Armeni, Gleb Donin, Francesco Costa.

**Supervision:** Patrizio Armeni, Francesco Costa.

**Visualization:** Ludovica Borsoi.

**Writing – original draft:** Ludovica Borsoi.

**Writing – review & editing:** Ludovica Borsoi, Patrizio Armeni, Gleb Donin, Francesco Costa, Luigi Ferini-Strambi.

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
