## [Decision Letter · Decision Letter 0]

13 Apr 2021

PONE-D-21-03912

The invisible costs of obstructive sleep apnea (OSA): a cost-of-illness analysis

PLOS ONE

Dear Dr. Borsoi,

Thank you for submitting your manuscript to PLOS ONE. After careful consideration, we feel that it has merit but does not fully meet PLOS ONE’s publication criteria as it currently stands. Therefore, we invite you to submit a revised version of the manuscript that addresses the points raised during the review process.

While two reviewers were quite positive with still some comments to be addressed, one reviewer was very critical. This reviewer has raised many important issues which I find of value and which I share. Therefore, we request that you provide a revision with all these issues addressed and possibly amended.

We look forward to receiving your revised manuscript.

Kind regards,

Thomas Penzel

Academic Editor

PLOS ONE

Journal Requirements:

We note that your literature search was performed on 2018;to allow an up-to-date view of the topic, we would request that the search is updated. Moreover, to meet our criteria on reproducibility, please provide more information on how data obtained from the systematic review was analysed and used for your cost- of-illness analysis.

We note that this manuscript is a systematic review or meta-analysis; our author guidelines therefore require that you use PRISMA guidance to help improve reporting quality of this type of study. Please upload copies of the completed PRISMA checklist as Supporting Information with a file name “PRISMA checklist”.

Thank you for stating the following in the Financial Disclosure section:

CERGAS SDA Bocconi received an unrestricted grant for research from Philips S.p.A. The funder had no role in study design, data collection and analysis, decision to publish, or preparation of the manuscript.

We note that you received funding from a commercial source: Philips S.p.A

Thank you for stating the following in the Competing Interests section:

Ludovica Borsoi, Patrizio Armeni, Gleb Donin and Francesco Costa have no competing interests to declare. Luigi Ferini-Strambi declares the following competing interests (last 3 years): Philips-Respironics (fee for lectures), Resmed (fee for advisory board).

Reviewers' comments:

Reviewer's Responses to Questions

**Comments to the Author**

1. Is the manuscript technically sound, and do the data support the conclusions?

Reviewer #1: Partly

Reviewer #2: Yes

Reviewer #3: Yes

2. Has the statistical analysis been performed appropriately and rigorously? 

Reviewer #1: Yes

Reviewer #2: Yes

Reviewer #3: Yes

3. Have the authors made all data underlying the findings in their manuscript fully available?

Reviewer #1: Yes

Reviewer #2: Yes

Reviewer #3: Yes

4. Is the manuscript presented in an intelligible fashion and written in standard English?

Reviewer #1: Yes

Reviewer #2: Yes

Reviewer #3: Yes

5. Review Comments to the Author

Reviewer #1: This study submitted by Borsoi and colleagues aimed at assessing the economic burden of OSA in the adult population in Italy. The attempt is interesting but there are several concerns regarding these data:

1-Litterature review is by far non exhaustive.

2-The novelty is limited as previous similar analysis and markov models have been conducted and published both in scientific journals or reported by Frost and Sullivan and Mc Kinsey (“The price of fatigue”) for the American academy of sleep medicine.

3-The main novelty is to provide these data for Italy. I think that the paper is more suitable for an Italian journal of health policy or management.

Reviewer #2: I think the topic is of outmost importance. The problem is hugely undermined in daily life. Therefore the possible outcomes in socioeconomic domain is a smart one to explore.

I only have one comment to the authors: We have seen in the last Covid outbreak that no tow country is same for medical measures. Therefore the sentence on line 136 "referred to other countries whose health care systems

can be comparable to the Italian one" may actually be off.

Reviewer #3: Obstructive sleep apnoea (OSA) is an underdiagnosed chronic disease with a high prevalence in adults. As it is becoming a significant social problem associated with a low quality of life and increased mortality, the cost-effectiveness ratio of diagnostic and therapeutic management of OSA is important to counteract the demand of objective diagnosis. This cost-of-illness study is considered an essential evaluation technique in health care, helping health-care decision-makers to set up and prioritize health-care policies and interventions. Therefore, this well written manuscript is worth being published.

6. PLOS authors have the option to publish the peer review history of their article (what does this mean?). If published, this will include your full peer review and any attached files.

Reviewer #1: No

Reviewer #2: No

Reviewer #3: **Yes: **Sophia E Schiza

---

## [Author Response · Author response to Decision Letter 0]

27 May 2021

Dear Academic Editor and Reviewers, 

We would like to thank you for your valuable comments, which helped us to further improve the manuscript. Please find below our answers to each revision request.

Academic Editor

We thank the Editor for this comment. We have checked the entire manuscript and revised the title, citing the method used (i.e. systematic review).

2. We note that your literature search was performed on 2018;to allow an up-to-date view of the topic, we would request that the search is updated. Moreover, to meet our criteria on reproducibility, please provide more information on how data obtained from the systematic review was analysed and used for your cost- of-illness analysis.

We thank the Editor for this useful comment. We have updated the review as requested (last update: 13th May 2021), revised the manuscript and all other materials (figures, supplementary files). Please, note that, besides the manuscript, the following files have been revised: Fig 1, Fig 2, Fig 3, S1 Table, S4 File, S5 File, S7 File. We reported the detail on how data obtained from the systematic review was analysed and used for cost-of-illness analysis in the main text. In particular, main information can be found in the methods section titled “Identification of conditions associated with OSA”. 

3. We note that this manuscript is a systematic review or meta-analysis; our author guidelines therefore require that you use PRISMA guidance to help improve reporting quality of this type of study. Please upload copies of the completed PRISMA checklist as Supporting Information with a file name “PRISMA checklist”.

We thank the Editor for the comment. The PRISMA checklist had been already included in the original submission as supporting file (S4 File). However, we have updated it after the changes made to the manuscript.

4. Thank you for stating the following in the Financial Disclosure section:

CERGAS SDA Bocconi received an unrestricted grant for research from Philips S.p.A. The funder had no role in study design, data collection and analysis, decision to publish, or preparation of the manuscript.

We note that you received funding from a commercial source: Philips S.p.A

Within this Competing Interests Statement, please confirm that this does not alter your adherence to all PLOS ONE policies on sharing data and materials by including the following statement: "This does not alter our adherence to PLOS ONE policies on sharing data and materials.”

We thank the Editor for the comment. We have changed the Competing Interests Statement and reported the amended version in the cover letter as requested.

Ludovica Borsoi, Patrizio Armeni, Gleb Donin and Francesco Costa have no competing interests todeclare. Luigi Ferini-Strambi declares the following competing interests (last 3 years): Philips-Respironics (fee for lectures), Resmed (fee for advisory board).

Please confirm that this does not alter your adherence to all PLOS ONE policies on sharing data and materials, by including the following statement: "This does not alter our adherence to PLOS ONE policies on sharing data and materials.” (as detailed online in our guide for authorshttp://journals.plos.org/plosone/s/competing-interests). If there are restrictions on sharing of data and/or materials, please state these. Please note that we cannot proceed with consideration of your article until this information has been declared.

We thank the Editor for the comment. We have changed the Competing Interests Statement and reported the amended version in the cover letter as requested.

Reviewer #1

This study submitted by Borsoi and colleagues aimed at assessing the economic burden of OSA in the adult population in Italy. The attempt is interesting but there are several concerns regarding these data:

1. Literature review is by far non exhaustive.

We thank the Reviewer for the comment. We acknowledge the fact that our inclusion criteria are particularly strict, since we decided to include only systematic reviews and meta-analyses. The reason for this choice is that we were looking for directional associations between OSA and other conditions and realized that in most cases the literature was extremely heterogeneous in terms of methods, definitions and results: therefore we opted for the “systematic review of systematic reviews” (Smith et al., 2011) which is a recommended method in this kind of cases. Moreover, we further restricted the selection on systematic reviews presenting a meta-analysis because we needed a reliable quantitative assessment of the intensity of association between OSA and other conditions. We know that a vast literature exists, suggesting that OSA might be connected to many conditions, but in some cases there is no sufficient quantitative evidence and in other cases there is no quantitative evidence at all. We decided to keep a conservative approach. Our results show that, even based only on the most reliable quantitative evidence (systematic reviews with meta-analyses) the economic impact of OSA on the society is huge, well beyond the one suggested so far by less structured studies. In addition, with respect to the original manuscript, we have updated the literature review to 2021 in order to collect relevant evidence published after the original search carried out on November 19th, 2018. We have acknowledged in the “Study limitations” section that the review performed may not be fully exhaustive as we did not consider individual studies but focused only on systematic reviews and meta-analyses, including the latter for COI analysis and included the methodological reference suggesting the “systematic review of systematic reviews” approach. In this perspective, the aim of our literature review is to be exhaustive in systematic reviews and meta-analyses only. 

2. The novelty is limited as previous similar analysis and markov models have been conducted and published both in scientific journals or reported by Frost and Sullivan and Mc Kinsey (“The price of fatigue”) for the American academy of sleep medicine.

3. The main novelty is to provide these data for Italy. I think that the paper is more suitable for an Italian journal of health policy or management.

We thank the Reviewer for these comments (2 and 3). We acknowledge the fact that other studies have been previously investigated the topic in other national contexts (we have cited the most relevant and comprehensive ones in the “Introduction” section). However, some of them either did not perform a systematic review to identify the conditions influenced by OSA (e.g., Hillman et al, 2018) or were not published in peer-reviewed journals (e.g., report by McKinsey) and in general all of them followed a less systematic and structured approach, thus not limiting the risk of biased analysis (e.g. the report by McKinsey was not transparent on the reason for inclusion of specific association measures between OSA and other conditions). We believe that, given the enormous amount of studies, but the relative scarcity of reliable synthetic quantitative information, a well structured, transparent and solid approach is necessary when investigating OSA’s relationship with other clinical and non-clinical conditions. Therefore, in addition to having adopted a systematic approach in the identification of studies reporting quantitative data on the influence of OSA on all other clinical and non-clinical conditions, the novelty of our study is that we included in our cost-of-illness analysis exclusively those providing the highest level of evidence, i.e. meta-analyses, overcoming the limitations of single studies. Moreover, unlike other studies, we provided also estimates on the QALYs value lost due to undertreatment of OSA. This is a major point, with strong policy implications, since it allows to measure not only the financial component of the cost-of-illness but also the economic counterpart of the health benefit lost by patients (which is the most important object of public expenditure in health). We show how important this component of the burden is, and no previous study did something similar. Although our results are focused on a specific national context (i.e., Italy), we believe that the methodological approach used, and extensively reported in our manuscript, may be interesting also for a non-Italian readership, also taking into account that all other studies are based on specific national contexts (e.g. Australia, the US, etc.). Differently from the previous literature, however, the specific context could be considered a limitation only with respect to epidemiological data, while our per-patient results are more generalizable than the ones presented in previous studies since we based our analysis on synthetic evidence and not on single studies.

Reviewer #2

1. I think the topic is of outmost importance. The problem is hugely undermined in daily life. Therefore the possible outcomes in socioeconomic domain is a smart one to explore. I only have one comment to the authors: We have seen in the last Covid outbreak that no tow country is same for medical measures. Therefore the sentence on line 136 "referred to other countries whose health care systems can be comparable to the Italian one" may actually be off.

We completely agree with the Reviewer’s point and revised the sentence. Moreover, we have also reported this limitation in a dedicated section (“Study limitations”). We thank the Reviewer for the useful comment. 

Reviewer #3

1. Obstructive sleep apnoea (OSA) is an underdiagnosed chronic disease with a high prevalence in adults. As it is becoming a significant social problem associated with a low quality of life and increased mortality, the cost-effectiveness ratio of diagnostic and therapeutic management of OSA is important to counteract the demand of objective diagnosis. This cost-of-illness study is considered an essential evaluation technique in health care, helping health-care decision-makers to set up and prioritize health-care policies and interventions. Therefore, this well written manuscript is worth being published.

We thank the Reviewer for the positive and very kind feedback.

---

## [Decision Letter · Decision Letter 1]

17 Jun 2021

PONE-D-21-03912R1

The invisible costs of obstructive sleep apnea (OSA): Systematic review and cost-of-illness analysis

PLOS ONE

Dear Dr. Borsoi,

Thank you for submitting your manuscript to PLOS ONE. After careful consideration, we have decided that your manuscript does not meet our criteria for publication and must therefore be rejected.

Specifically:

Based on the reviewer comments, which were quite different and based on the diversity of comments, in conclusion, we like to follow the more critical decision. May be a more specialized journal will appreciate more your specific work.

I am sorry that we cannot be more positive on this occasion, but hope that you appreciate the reasons for this decision.

Yours sincerely,

Thomas Penzel

Academic Editor

PLOS ONE

Additional Editor Comments (if provided):

Dear authors, we acknowledge the large amount of data. This is exceptional. We also acknowledge the hard work provided. However regarding a cost-of-illness analysis, our knowledgeable reviewers remain to be very critical. Therefore we decided to reject the manuscript.

Reviewers' comments:

Reviewer's Responses to Questions

**Comments to the Author**

1. If the authors have adequately addressed your comments raised in a previous round of review and you feel that this manuscript is now acceptable for publication, you may indicate that here to bypass the “Comments to the Author” section, enter your conflict of interest statement in the “Confidential to Editor” section, and submit your "Accept" recommendation.

Reviewer #4: (No Response)

2. Is the manuscript technically sound, and do the data support the conclusions?

Reviewer #4: Partly

3. Has the statistical analysis been performed appropriately and rigorously? 

Reviewer #4: Yes

4. Have the authors made all data underlying the findings in their manuscript fully available?

Reviewer #4: Yes

5. Is the manuscript presented in an intelligible fashion and written in standard English?

Reviewer #4: Yes

6. Review Comments to the Author

Reviewer #4: Enormous amount of data, and hardworking, sophisticated approach to a very complex topic, revealing many levels of socio-economic approach to health economy.

Well realized literature update, text, and data ameliorations according to the first couple of reviews!

The country preference (Italy) in view of this reviewer seems of minor relevance, because data mainly rely on costs, and prevalence of diverse illnesses in relation to OSA. The latter are rather independant of country.

Thus the elaborate methods are developed originally, and compiled in a concise way, presented in relatively brief descriptions.

The presentation of results mainly rely on the population attributable fraction (PAF), which is enhanced by use of numerous specific variables of health economy (OR, RR, HR, QALY, WTP, and different measures of OSA-probability). The reason for this elaborated development of PAF remains unclear (in the methods description). Moreover it is not discussed or critically reflected in the discussion/conclusions part. These methods are an innovative approach to the complex topic of compiling the "invisible" (indirect) costs of OSA. Because of reasons of this intelligent, unique approach to specific topics of socio-economy, it becomes clear that the article does not completely fulfill the criteria of PLOS.

A journal of health economy seems appropriate for publication (e.g. Eur. J Health Econ., Health Economics, J Health Economics).

Their readers are suggested to be customized to the methodological approach.

Thus reviewer encourages the authors to publish in one of these lines of business.

With respect to the many additional S1-9 files it seems rather difficult to understand data processing completely.

In relation to this, the length of the manuscript is rather brief, but overcrowded by the many assumptions (partly from literature) to implement the various items and methodological techniques of health economy.

Question: Female sexual dysfunction for development of PAF data in OSA is well accepted. But what is the contribution of Caesarean delivery to PAF in OSA?

General discussion: Aren't the variables used rather part of indirect costs instead of invisible (intangible) costs used in the title?

7. PLOS authors have the option to publish the peer review history of their article (what does this mean?). If published, this will include your full peer review and any attached files.

Reviewer #4: No

- - - - -

---

## [Author Response · Author response to Decision Letter 1]

15 Feb 2022

Dear Academic Editor and Reviewer, 

Please find below our answers to your valuable comments.

Academic Editor

1. Based on the reviewer comments, which were quite different and based on the diversity of comments, in conclusion, we like to follow the more critical decision. May be a more specialized journal will appreciate more your specific work.

I am sorry that we cannot be more positive on this occasion, but hope that you appreciate the reasons for this decision.

Additional Editor Comments :

Dear authors, we acknowledge the large amount of data. This is exceptional. We also acknowledge the hard work provided. However regarding a cost-of-illness analysis, our knowledgeable reviewers remain to be very critical. Therefore we decided to reject the manuscript.

The Academic Editor's decision appears to be in contrast with the comments received by the majority of reviewers in both rounds of revision and is also manifestly inconsistent with the points on which the Editor asked the authors to intervene on. In fact, in the first round of revision (mainly favorable, as two of the reviewers outlined the importance of the topic, and one of them explicitly underlined that the manuscript was worth being published), the Editor asked us to update the systematic review in order to "allow an up-to-date view of the topic". Following the Editors' comment, we updated the literature review, that required a substantial work. However, in the second round of revision, despite having acknowledged "the large amount of data", "the hard work provided" and the rigor of the methods used (reviewer's comment: "Enormous amount of data, and hardworking, sophisticated approach to a very complex topic", "Well realized literature update, text, and data ameliorations", "methods are developed originally, and compiled in a concise way"), the Editor decided that the cost-of-illness was not of interest of the journal ("a more specialized journal will appreciate more your specific work"). Moreover, we would like to underline that PLOS ONE has published several cost-of-illness studies over the last years, and some of them have used methods similar to those presented in our paper. We include here a non-exhaustive but exemplary list of cost-of-illness analyses published on the journal, some of which are very recent:

• Armour, Mike, et al. "The cost of illness and economic burden of endometriosis and chronic pelvic pain in Australia: A national online survey." PloS one 14.10 (2019): e0223316.

• Curado, Daniel da Silva Pereira, et al. "Direct cost of systemic arterial hypertension and its complications in the circulatory system from the perspective of the Brazilian public health system in 2019." PloS one 16.6 (2021): e0253063.

• de Oliveira, Michele Lessa, Leonor Maria Pacheco Santos, and Everton Nunes da Silva. "Direct healthcare cost of obesity in Brazil: an application of the cost-of-illness method from the perspective of the public health system in 2011." PloS one 10.4 (2015): e0121160.

• Ernstsson, Olivia, et al. "Cost of illness of multiple sclerosis-a systematic review." PloS one 11.7 (2016): e0159129.

• Ilboudo, Patrick G., et al. "Cost-of-illness of cholera to households and health facilities in rural Malawi." PloS one 12.9 (2017): e0185041.

• Javanbakht, Mehdi, et al. "Cost-of-illness analysis of type 2 diabetes mellitus in Iran." PloS one 6.10 (2011): e26864.

• Jo, Minkyung, et al. "The cost-of-illness trend of schizophrenia in South Korea from 2006 to 2016." PloS one 15.7 (2020): e0235736.

• Matsumoto, Kunichika, et al. "Cost of illness of hepatocellular carcinoma in Japan: A time trend and future projections." Plos one 13.6 (2018): e0199188.

• Moran, Patrick S., et al. "Economic burden of maternal morbidity–A systematic review of cost-of-illness studies." PloS one 15.1 (2020): e0227377.

• Oliveira, Luana Nice da Silva, Alexander Itria, and Erika Coutinho Lima. "Cost of illness and program of dengue: A systematic review." PloS one 14.2 (2019): e0211401.

• Pares-Badell, Oleguer, et al. "Cost of disorders of the brain in Spain." PloS one 9.8 (2014): e105471.

• Steinke, Sabine IB, et al. "Cost-of-illness in psoriasis: comparing inpatient and outpatient therapy." PLoS One 8.10 (2013): e78152.

Based on the positive comments on the methodological quality and relevance of results and on the fact that this kind of studies has been often published in this journal, we believe that this manuscript could be relevant for PLOS ONE.

Reviewer #4: 

1. Enormous amount of data, and hardworking, sophisticated approach to a very complex topic, revealing many levels of socio-economic approach to health economy.

Well realized literature update, text, and data ameliorations according to the first couple of reviews!

The country preference (Italy) in view of this reviewer seems of minor relevance, because data mainly rely on costs, and prevalence of diverse illnesses in relation to OSA. The latter are rather independent of country.

Thus the elaborate methods are developed originally, and compiled in a concise way, presented in relatively brief descriptions.

We thank the reviewer for the positive feedback.

2. The presentation of results mainly rely on the population attributable fraction (PAF), which is enhanced by use of numerous specific variables of health economy (OR, RR, HR, QALY, WTP, and different measures of OSA-probability). The reason for this elaborated development of PAF remains unclear (in the methods description). Moreover it is not discussed or critically reflected in the discussion/conclusions part. 

These methods are an innovative approach to the complex topic of compiling the "invisible" (indirect) costs of OSA. Because of reasons of this intelligent, unique approach to specific topics of socio-economy, it becomes clear that the article does not completely fulfill the criteria of PLOS.

A journal of health economy seems appropriate for publication (e.g. Eur. J Health Econ., Health Economics, J Health Economics).

Their readers are suggested to be customized to the methodological approach.

Thus reviewer encourages the authors to publish in one of these lines of business.

We thank the reviewer for the comments. As regards the PAF, as explained in the methods section, we had to rely on different formulae as the measures of association of OSA with other conditions found in the meta-analyses included were heterogeneous (i.e., relative risk, odds ratio or hazard ratio) and required different approaches. In the manuscript, we have clearly specified the references we relied on in order to select the most appropriate formulae on the basis of association data. 

As regards the adherence of our manuscript with the journal's scope, we would like to underline that PLOS ONE has published several cost-of-illness studies over the last years, and some of them have used methods similar to those presented in our paper. Therefore, we believe that our manuscript could be relevant for the journal and its audience. Please find below some examples of cost-of-illness studies published on PLOS ONE:

• Armour, Mike, et al. "The cost of illness and economic burden of endometriosis and chronic pelvic pain in Australia: A national online survey." PloS one 14.10 (2019): e0223316.

• Curado, Daniel da Silva Pereira, et al. "Direct cost of systemic arterial hypertension and its complications in the circulatory system from the perspective of the Brazilian public health system in 2019." PloS one 16.6 (2021): e0253063.

• de Oliveira, Michele Lessa, Leonor Maria Pacheco Santos, and Everton Nunes da Silva. "Direct healthcare cost of obesity in Brazil: an application of the cost-of-illness method from the perspective of the public health system in 2011." PloS one 10.4 (2015): e0121160.

• Ernstsson, Olivia, et al. "Cost of illness of multiple sclerosis-a systematic review." PloS one 11.7 (2016): e0159129.

• Ilboudo, Patrick G., et al. "Cost-of-illness of cholera to households and health facilities in rural Malawi." PloS one 12.9 (2017): e0185041.

• Javanbakht, Mehdi, et al. "Cost-of-illness analysis of type 2 diabetes mellitus in Iran." PloS one 6.10 (2011): e26864.

• Jo, Minkyung, et al. "The cost-of-illness trend of schizophrenia in South Korea from 2006 to 2016." PloS one 15.7 (2020): e0235736.

• Matsumoto, Kunichika, et al. "Cost of illness of hepatocellular carcinoma in Japan: A time trend and future projections." Plos one 13.6 (2018): e0199188.

• Moran, Patrick S., et al. "Economic burden of maternal morbidity–A systematic review of cost-of-illness studies." PloS one 15.1 (2020): e0227377.

• Oliveira, Luana Nice da Silva, Alexander Itria, and Erika Coutinho Lima. "Cost of illness and program of dengue: A systematic review." PloS one 14.2 (2019): e0211401.

• Pares-Badell, Oleguer, et al. "Cost of disorders of the brain in Spain." PloS one 9.8 (2014): e105471.

• Steinke, Sabine IB, et al. "Cost-of-illness in psoriasis: comparing inpatient and outpatient therapy." PLoS One 8.10 (2013): e78152.

3. With respect to the many additional S1-9 files it seems rather difficult to understand data processing completely.

In relation to this, the length of the manuscript is rather brief, but overcrowded by the many assumptions (partly from literature) to implement the various items and methodological techniques of health economy.

We thank the reviewer for the comments. The structure of our manuscript is the result of a clear and reasoned choice. In the main text, indeed, we provide information on the most relevant methodological aspects that allow the reader to understand the results presented. Through the supplementary materials, instead, we aimed at providing additional information on some methodological steps that, despite not being fundamental to understand the concepts discussed in the main text, are deemed useful in order to fully grasp the results obtained and ensure transparency of our work. 

We acknowledge that we relied on some assumptions in order to develop our analyses, but we thoroughly justified and referenced them throughout the text.

4. Female sexual dysfunction for development of PAF data in OSA is well accepted. But what is the contribution of Caesarean delivery to PAF in OSA?

We thank the reviewer for the question. There are several studies, included in the meta-analysis by Liu et al (2019), which demonstrated that OSA is related to an increased risk for Caesarean delivery. Therefore, as our study aims at estimating the economic burden influenced by OSA, i.e. the proportion of the cost-of-illness of conditions for which OSA is a proven risk factor that can be attributed to OSA itself, we included this maternal-related outcome in our analyses.

5. General discussion: Aren't the variables used rather part of indirect costs instead of invisible (intangible) costs used in the title?

We thank the reviewer for the question. When we talk about the "invisible costs" of OSA in the title, we are not referring to a specific cost category (e.g. indirect costs), but rather to the fact that the burden generated by OSA is somehow hidden (thus "invisible") behind the cost of other conditions for which it is a risk factor.

---

## [Decision Letter · Decision Letter 2]

6 May 2022

The invisible costs of obstructive sleep apnea (OSA): Systematic review and cost-of-illness analysis

PONE-D-21-03912R2

Dear Dr. Borsoi,

We’re pleased to inform you that your manuscript has been judged scientifically suitable for publication and will be formally accepted for publication once it meets all outstanding technical requirements.

Kind regards,

Tai-Heng Chen, M.D.

Academic Editor

PLOS ONE

Reviewers' comments:

Reviewer's Responses to Questions

**Comments to the Author**

1. If the authors have adequately addressed your comments raised in a previous round of review and you feel that this manuscript is now acceptable for publication, you may indicate that here to bypass the “Comments to the Author” section, enter your conflict of interest statement in the “Confidential to Editor” section, and submit your "Accept" recommendation.

Reviewer #4: All comments have been addressed

Reviewer #5: All comments have been addressed

Reviewer #6: All comments have been addressed

2. Is the manuscript technically sound, and do the data support the conclusions?

Reviewer #4: Yes

Reviewer #5: Yes

Reviewer #6: Yes

3. Has the statistical analysis been performed appropriately and rigorously? 

Reviewer #4: Yes

Reviewer #5: Yes

Reviewer #6: Yes

4. Have the authors made all data underlying the findings in their manuscript fully available?

Reviewer #4: Yes

Reviewer #5: Yes

Reviewer #6: Yes

5. Is the manuscript presented in an intelligible fashion and written in standard English?

Reviewer #4: Yes

Reviewer #5: Yes

Reviewer #6: Yes

6. Review Comments to the Author

Reviewer #4: The value of this review is the approach to give numbers to invisible health care costs, especially the consideration in relation to various diseases!

But things for CPAP may change very fast: preferred mode of therapy, additional kinds of other therapies, patient's adherence, national healthcare conditions, etc. This will have enormous impact on costs. Could you briefly refer to such developments?

Reviewer #5: The authors have well answered to most of queries.

This is an important topic to consider in beween medecine and economics.

Reviewer #6: After the previous revisions of the text, I suggest that the manuscript is ready for the publication.

I think it is interesting for the Plos One Journal.

7. PLOS authors have the option to publish the peer review history of their article (what does this mean?). If published, this will include your full peer review and any attached files.

Reviewer #4: No

Reviewer #5: No

Reviewer #6: No

---

## [Editor Report · Acceptance letter]

12 May 2022

PONE-D-21-03912R2 

The invisible costs of obstructive sleep apnea (OSA): Systematic review and cost-of-illness analysis 

Dear Dr. Borsoi:

I'm pleased to inform you that your manuscript has been deemed suitable for publication in PLOS ONE. Congratulations! Your manuscript is now with our production department. 

Kind regards, 

on behalf of

Dr. Tai-Heng Chen 

Academic Editor

PLOS ONE